# Pioglitazone-Primed Mesenchymal Stem Cells Stimulate Cell Proliferation, Collagen Synthesis and Matrix Gene Expression in Tenocytes

**DOI:** 10.3390/ijms20030472

**Published:** 2019-01-22

**Authors:** Won Kim, Seul Ki Lee, Young-Won Kwon, Sun G. Chung, Soo Kim

**Affiliations:** 1Department of Rehabilitation Medicine, Asan Medical Center, University of Ulsan College of Medicine, Seoul 05505, Korea; duocl79@gmail.com (W.K.); kywon012@naver.com (Y.-W.K.); 2Department of Rehabilitation Medicine, College of Medicine, Seoul National University, 101 Daehak-ro, Jongno-gu, Seoul 03080, Korea; 3Stem Cell Center, Asan Institute for Life Science, Asan Medical Center, Seoul 05505, Korea; clotilda75@gmail.com; 4Institute of Aging, Seoul National University, Seoul 03080, Korea; 5Rheumatism Research Institute, Medical Research Center, Seoul National University, Seoul 03080, Korea

**Keywords:** MSC, priming, pioglitazone, tenocyte, regeneration

## Abstract

Various therapeutic effects of mesenchymal stem cells (MSCs) have been reported. However, the rapid clearance of these cells in vivo, difficulties in identifying their therapeutic mechanism of action, and insufficient production levels remain to be resolved. We investigated whether a pioglitazone pre-treatment of MSCs (Pio-MSCs) would stimulate the proliferation of co-cultured tenocytes. Pioglitazone increased the proliferation of MSCs and enhanced the secretion of VEGF (vascular endothelial growth factor) and collagen in these cells. We then examined the effects of Pio-MSCs on tenocytes using an indirect transwell culture system. A significant increase in tenocyte proliferation and cell cycle progression was observed in these co-cultures. Significant increases were observed in wound scratch closure by tenocytes from a Pio-MSC co-culture. Pio-MSCs also enhanced the secretion of collagen from tenocytes. A higher mRNA level of collagen type 1 (Col 1) and type 3 (Col 3), scleraxis (Scx), and tenascin C (TnC) was found in the tenocytes in Pio-MSC co-cultures compared with monocultured cells or tenocytes cultured with non-treated MSCs. Our results indicate that pioglitazone enhances the therapeutic effects of MSCs on tendon repair.

## 1. Introduction

Tendon injuries are one of the most common musculoskeletal problems treated by clinicians [1]. Overuse and age-related degeneration is the main causes of these injuries as they can result in tendinosis and a progression to tendon tearing. Unfortunately, the natural healing ability of tendons is very poor due to their low metabolic rate, hypocellularity, and hypovascularity [2,3]. Treatment outcomes for chronic tendinopathy are therefore often unsatisfactory and retearing is common after surgical repair [4]. 

Previous animal studies have reported relatively favorable results from stem cell therapy for tendon injuries [5], although other studies have reported negative results from this approach [2]. There is however little available evidence on the outcomes of stem cell therapy for tendon injuries from well-designed human studies [6]. Several effect enhancement methods, such as genetic modification and growth factors co-delivery, have therefore been continuously attempted [2,7,8]. 

Mesenchymal stem cells (MSCs) are progenitor cells of connective tissues (osteoblasts, chondrocytes, and adipocytes) with significant immunoregulatory and regenerative functions [9]. Despite their potential for use in therapeutic applications, there are several hurdles to this such as the low viability of the transplanted cells, innate heterogenicity, and unidentified factors related to aging of the donor.

Pioglitazone is one of the currently popular hypoglycemic agents acting on peroxisome proliferator-activated receptor gamma (PPAR-γ) [10]. PPAR-γ activation seems to affect the differentiation of MSCs [11,12]. Recent studies have also reported that pioglitazone-pretreated MSCs showed significantly improved therapeutic effects in myocardial infarction and emphysema animal models [12,13].

The aim of our present study was to investigate the effects of pioglitazone-pretreated MSCs derived from human Wharton’s jelly (Pio-MSCs) on tenocytes using an indirect transwell culture system. We found that pioglitazone enhances the therapeutic effects of MSCs on tendon repair, as shown by the increased proliferation, migration, matrix gene expression, and collagen secretion of tenocytes.

## 2. Results

### 2.1. Identification of MSCs and Pioglitazone-Pretreated MSCs by Morphology and Flow Cytometry 

We sought to investigate whether pioglitazone-pretreated MSCs (Pio-MSCs) can stimulate the proliferation of the cells for tendon regeneration, i.e., tenocytes. We first isolated and cultured MSCs from the human Wharton’s jelly (WJ) of neonatal umbilical cord tissue and then cultured these cells until passage 5 (Figure 1). Throughout this present study, we used three different WJ-MSC lines, each of which was pretreated with pioglitazone and then compared in terms of characteristics and functionality as an MSC. No morphological differences between MSCs and Pio-MSCs were found. Flow cytometric analysis revealed that Pio-MSCs were positive for typical MSC surface markers such as CD73, CD90, and CD105, and negative for CD34. 

### 2.2. Pioglitazone Promotes the Proliferation and Soluble Protein Secretion of MSCs

To identify the effects of pioglitazone pre-treatment on MSCs, we compared the proliferation and soluble ECM (extracellular matrix) protein secretion of these cells with that of untreated MSCs. As shown in Figure 2a, Pio-MSC proliferation was significantly increased compared with untreated MSCs at day 7 (*p* = 0.037), whereas no difference was found on days 1 or 4. We next evaluated whether pioglitazone stimulates the secretion of collagen and also VEGF (vascular endothelial growth factor), which is a major regeneration mediator protein secreted by these cells. Figure 2b shows that Pio-MSCs had an enhanced level of VEGF secretion compared with the untreated cells (*p* < 0.05). Figure 2c indicates that an increase in soluble collagen level was observed in Pio-MSCs (*p* = 0.07), although this was not significantly different from MSCs.

### 2.3. Assessment of the Role of Pio-MSCs in Tenocyte Proliferation 

To study the impact of co-culturing tenocytes with MSCs and Pio-MSCs, co-culture groups and control groups were established as shown in Figure 3a. A transwell co-culture system was established using six-well plates with equivalent numbers of tenocytes, tenocytes with MSCs, and tenocytes with Pio-MSCs. We then investigated whether the MSC and Pio-MSC co-cultures stimulated the growth of tenocytes. As shown in Figure 3b, a 3-(4,5-dimethylthiazol-2-yl)-2,5-diphenyltetrazolium bromide (MTT) assay revealed that Pio-MSCs significantly increased the proliferation of tenocytes. We thus conducted cell cycle analysis to confirm the proliferative role of Pio-MSCs in this tenocyte co-culture system. As shown in Figure 3c, a co-culture of Pio-MSCs and tenocytes increased the number of cells in S phase as compared with tenocytes alone or an MSC/tenocyte co-culture. However, this was not statistically significant.

### 2.4. Tenocyte Migration Assay

A migration assay revealed a significantly increased migration area for tenocytes following co-culture with MSCs or Pio-MSCs using an indirect co-culture system, as compared with a tenocyte only group at 6 h, 12 h, and 24 h (Figure 4). 

### 2.5. Soluble Collagen Protein and mRNA Expression Analysis

We next determined whether Pio-MSCs stimulated the secretion of soluble collagen from tenocytes, as this is a critical wound healing mediator [14]. As shown in Figure 5, a significant increase in collagen secretion was evident in the Pio-MSC/tenocyte co-culture group (*p* < 0.05) compared with the tenocytes only group and the MSC/tenocyte co-culture group. To determine whether Pio-MSCs regulate the expression of genes related to tendon regeneration, tenocytes co-cultured with MSCs and Pio-MSCs were analyzed using a quantitative reverse-transcription polymerase chain reaction (qRT-PCR). Genes involved in wound healing (collagen type 1 (Col 1), collagen type 3 (Col 3), Scleraxis (Scx), tenascin C (TnC)) were indeed found to have a significantly increased expression in tenocytes co-cultured with Pio-MSCs (Figure 6, *p* < 0.05). 

### 2.6. Immunoblotting and Animal Experiments

We investigated whether the proliferation of Pio-MSCs was mediated through the activation of protein kinase B (AKT) and extracellular signal-regulated kinase (ERK)-1/2, which are known to be important for cell proliferation [15,16]. No difference on the phosphorylation of AKT or ERK was found between tenocytes culture with MSCs or Pio-MSCs (Appendix A). Also we detected VEGF, collagen I, and collagen III in the rat tendon injected with MSCs or Pio-MSCs to investigate the therapeutic effects of MSCs and Pio-MSCs. The expression of VEGF, collagen I, and collagen III was increased in the Pio-MSC injected group and the expression of collagen I was significantly increased. (Appendix A)

## 3. Discussion

The main purpose of our present study was to explore whether pioglitazone-pretreated MSCs isolated and cultured from the Wharton’s jelly tissue of the neonatal umbilical cord (Pio-MSCs) could enhance the proliferation of tenocytes. No remarkable differences were detected between the characteristics of the same donor-derived basic MSCs and Pio-MSCs, including their morphology and cell surface marker expression profiles. Pioglitazone was found to promote the proliferation and the VEGF and collagen secretion of MSCs. We speculated that the increased secretion of VEGF from Pio-MSCs would be beneficial for angiogenesis at the tendons, which have poor vascularity [17]. We also surmised that this may enhance the regeneration function of tenocytes as it has been reported that VEGF enhances proliferation and the expression of tendon-related genes in human tenocytes [18]. We thus believed that Pio-MSCs would have a more beneficial impact than untreated MSCs in the treatment of an injured tendon.

Prior animal studies have reported the positive effects of MSCs therapy on tendon injury as a result of enhanced primary tendon healing and promotion of tissue repair via paracrine effects [5,19] However, the regained tendon strength was only 20–60% of that of a normal tendon [2]. In addition, because of the relatively small area involved with an injured tendon and the requirement for scaffolds, the delivery of a sufficient number of MSCs is sometimes difficult for tendon treatment. Increasing the potency and therapeutic effects of MSCs would therefore be necessary to treat an injured tendon in clinical practice. Co-localization, transgenic, and biochemical and priming/preconditioning approaches have been described to achieve this [20,21,22]. Among the methods for enhancing the potency of MSCs, we have found that pre-treatment with pioglitazone may be particularly useful for treating an injured tendon. Importantly also, priming stem cells with an approved drug such as pioglitazone has fewer safety and ethical issues than other enhancement methods such as genetic modification [7].

Pioglitazone is a PPAR-γ agonist that is widely used as a hypoglycemic agent for diabetes [23]. Several studies have found that pioglitazone facilitates the trans-differentiation of MSCs and improves their therapeutic effects. Shinmura et al. reported the pioglitazone induces the cardiomyogenic trans-differentiation of BM-MSCs via PPAR-γ activation and that the transplantation of pioglitazone pretreated MSCs improves cardiac function in a myocardial infarction animal model [12]. Hou et al. also demonstrated that Pioglitazone-treated MSCs improved cardiac function in a myocardial infarction model [24]. Hong et al. reported that pioglitazone-pretreated adipose derived stem cells had improved therapeutic effects in an emphysema mouse model [13]. These authors suggested that paracrine effects may be an important mechanism by which pioglitazone exerts its impact. The findings of these previous studies are consistent with our current observations that pioglitazone pre-treatment enhances the paracrine effects of MSCs.

We confirmed these effects using an indirect co-culture system with tenocytes and either MSCs or Pio-MSCs. Pio-MSCs accelerate the growth, migration, survival, and cell cycle progression of tenocytes. Pio-MSC/tenocyte co-cultures had more potent effects on the proliferation of tenocytes, and showed better growth kinetics, than the untreated MSC/tenocyte co-culture group or the tenocyte-only control group. In addition, an indirect co-culture with Pio-MSCs enhanced the tendon regeneration-related function of tenocytes. Collagen is the main structural protein in tendons, comprising 85% of the tendon dry weight [25]. Tenascin C and scleraxis participate in collagen fiber alignment and early tendon development, respectively [26]. Pio-MSCs enhanced collagen secretion, and the expression of tendon-related genes, such as Col 1, Col 3, SCX, and TnC, from tenocytes. These aforementioned effects of Pio-MSCs are likely to enhance tendon regeneration, compared with untreated MSCs.

To enhance their therapeutic potency, there have been various attempts to improve stem cell functions such as growth in hypoxic cultures, growth factor stimulation, and genetic manipulation. However, there are issues with these approaches in terms of demonstrating the safety and efficacy required for the development of these cells as therapeutic agents. Pioglitazone pre-treatment represented a method of enhancing the function of MSCs by using an already safety-proven drug. Importantly, there are no ethical problems with this approach compared with a transgenic method of enhancing stem cell function. In addition to its proven safety as a clinical agent, we have identified the enhancing effect of pioglitazone on MSC function, which has the potential to overcome a variety of difficulties in the commercialization of these stem cells as a viable treatment application. We confirmed that pioglitazone-primed MSCs have enhanced functional effects in terms of promoting the regenerative ability of tenocytes in a co-culture environment.

The use of pioglitazone-treated MSCs also has advantages over methods such as hypoxic culturing of MSCs or current medical practices on treating tenocytes. Moreover, the current protocols for the stimulation of tenocytes with autologous or allogenic stem cells are clearly suboptimal due to their low therapeutic outcomes in vivo [2,27]. It is known that the regenerative capacity of tendons is low, and that transplantation procedures are difficult due to the distinct anatomical structures of these tissues. Our current method could potentially increase the therapeutic potential of autologous stem cells collected from elderly patients, in which cellular activity is known to be significantly decreased. When autologous stem cells are used for tendon repair in aged individuals, various possibilities to correct their endogenous deficits or to pre-activate them ex vivo or in situ via growth factor stimulation or gene therapy, have to be carefully considered [2]. It is widely accepted that the function of MSCs are attributable to their paracrine nature [28] and we suggest from our current findings that Pio-MSCs can safely enhance the function of tenocytes.

Our current study is the first to confirm the feasibility of tendon regeneration therapy using MSCs treated with pioglitazone. There are, however, some limitations to our experiments of note. First, since most of the effects we describe were identified in vitro, animal studies using appropriate tendon disease models (e.g., tendinitis, tenosynovitis, etc.) are needed to confirm the efficacy of pioglitazone in priming MSCs. Second, we did not investigate these therapeutic effects in human tenocytes from older patients who would be most likely subjects for novel treatments for tendon injury. To overcome these limitations in the future, we will additionally examine the tendon regeneration effect of pre-treatment MSCs of pioglitazone using a tendon defect animal model, and test whether the pioglitazone priming effect on MSCs is different between young and aged donors.

## 4. Materials and Methods 

### 4.1. Isolation of Human MSCs and Culture of MSCs and Pio-MSCs

Wharton’s jelly MSCs were collected from human umbilical cords after a full-term, healthy delivery. The procedures for tissue harvesting and obtaining informed consent were approved by the Asan Medical Center Institutional Review Board (AMC IRB File No. 2015-3030, Approved on 2 April 2015). Written informed consent was obtained from all participants. The umbilical cords were washed several times with PBS prior to cutting them into small pieces (0.5–1 cm in length). Each piece was then longitudinally cut for the removal of all blood vessels. The matrix was scraped, minced into small pieces, and transferred to 100-mm tissue culture dishes (SPL Life Science, Pocheon-si, Korea). The cells were cultured for 7 days in minimum essential medium-α (MEM-α) (Thermo Fisher Scientific, Waltham, MA, USA) supplemented with 10% fetal bovine serum (FBS) and 1% Pen-Strep (Thermo Fisher Scientific, Waltham, MA, USA) at 37 °C in 5% CO_2_ and 95% humidified air. The culture medium was changed every 4 days. Upon reaching 80–90% confluency, cells were detached with TryPLE Express (Thermo Fisher Scientific, Waltham, MA, USA) and replated into culture flasks at a split ratio of 1:5. For Pio-MSCs, the MSC culture medium was treated with 3 μmol/ L pioglitazone for 7 days in MEM-α (Thermo Fisher Scientific, Waltham, MA, USA) supplemented with 10% FBS and 1% Pen-Strep (Thermo Fisher Scientific, Waltham, MA, USA) at 37 °C in 5% CO_2_.

### 4.2. Characterization of MSCs and Pio-MSCs 

For flow cytometry analysis, MSCs or Pio-MSCs were trypsinized and washed twice prior to resuspension in PBS containing 2% FBS and 1 mM EDTA. Cells were adjusted to 1 × 10^6^ in 100 µL of cell suspension. For cell surface labeling, cell suspensions were incubated at 4 °C for 30 min with 5 µL of antibodies (dilution, 1:20) against MSC-specific surface markers. Phycoerythrin (PE)-conjugated mouse anti-human CD73, fluorescein isothiocyanate (FITC)-conjugated mouse anti-human CD90, and PE-conjugated mouse anti-human CD105 antibodies were supplied by BD Biosciences (San Jose, CA, USA). FITC-conjugated mouse anti-human CD34 was supplied by BD PharMingenTM (San Jose, CA, USA). Cell surface marker analysis was performed using a BD FACSCanto™ II Flow Cytometer and FACSDIVA software version 6.1.3 (BD Biosciences, San Jose, CA, USA). 

### 4.3. Cell Proliferation and Relative Soluble Protein Secretion Analyses of MSCs and Pio-MSCs

The proliferation of MSCs and Pio-MSCs was determined using a standard MTT assay. Cells (1 × 10^4^ cells/well) were overnight cultured in 96-well plates and treated with 10 μL of a MTT reagent. A total of 100 μL of purple formazan crystals (R&D system, Minneapolis, MN, USA) were dissolved in detergent and added to the plate. The absorbance was recorded on a microplate reader at a wavelength of 570 nm. MSCs and Pio-MSCs (1 × 10^5^) were seeded in six-well plates and incubated at 37 °C in 5% CO_2_ to allow cell attachment. Cells were harvested and the concentration of VEGF and collagen in the supernatant was measured using a Human Magnetic Luminex^®^ Screening Assay (R&D Systems, Minneapolis, MN, USA) and SirCol assay kit (Biocolor Inc., Carrickfergus, UK), respectively.

### 4.4. Primary Culture of Rat Tenocytes

Achilles tendons were obtained from 9-week-old, male, healthy Sprague Dawley rats, cut into 2–3 mm^3^ pieces, and placed into six-well culture plates. Cells were propagated in Dulbecco’s modified Eagle’s medium (DMEM) with 10% FBS (Hyclone) in a humidified 5% CO_2_ incubator at 37 °C. After 5 days, the tenocytes were trypsinized and subcultured in 100 mm culture plates and the medium was changed every three days. Cells from passage 3 to 5 were used in this study. All study procedures received approval from the Institutional Animal Care and Use Committee of Asan Medical Center, Asan Institute for Life Sciences (IACUC Approval No.: 2018-12-006).

### 4.5. Tenocyte Proliferation, Cell Cycle, and Migration Assays

MSCs and Pio-MSCs were co-cultured with tenocytes in vitro using transwell chambers (BD Biosciences, San Jose, CA, USA). The transwell co-culture system was established in six-well plates with an equivalent number (5 × 10^4^) of tenocytes, tenocytes with MSCs, and tenocytes with Pio-MSCs. MSCs and Pio-MSCs were seeded in the inserts at a density of 5 × 10^4^ cells/insert and maintained in a humidified incubator with 5% CO_2_ at 37 °C. The proliferation of tenocytes was determined using a standard MTT assay. Tenocytes (1 × 10^4^ cells/well) were cultured overnight in 96-well plates and treated with 10 μL of MTT reagent (R&D system, Minneapolis, MN, USA). A total of 100 μL of purple formazan crystals were dissolved in detergent reagent added to the plate. The absorbance was recorded on a microplate reader at a wavelength of 570 nm. For cell cycle analysis, the tenocytes were stained with 1 μM of Vybrant™ DyeCycle™ Green according to manufacturer’s instructions and analyzed with a FACS Canto flow cytometer (BD Biosciences, Piscataway, NJ, USA). For the migration assay, tenocytes were cultured overnight at a density of 2 × 10^5^ cells/well in 6-well culture plates. Wounds were made by scratching using a sterile 1000 μL pipette tip. MSCs and Pio-MSCs were seeded in the inserts at a density of 1 × 10^5^ cells/insert and maintained in a humidified incubator with 5% CO_2_ at 37 °C. MSCs and Pio-MSCs were co-cultured with tenocytes using transwell chambers, after washing the tenocytes with PBS, images were obtained at 0 h, 6 h, 12 h, and 24 h using a phase-contrast microscope. The area of migration was quantified using ImageJ software (NIH, Bethesda, MD, USA) and normalized against the wound area at 0 h.

### 4.6. Soluble Collagen Analysis and Quantitative Real-Time PCR

Tenocytes (5 × 10^4^) were seeded into six-well plates and incubated at 37 °C to allow cell attachment. MSCs and Pio-MSCs were co-cultured with tenocytes in vitro using transwell chambers (BD Falcon). The transwell co-culture system was established in six-well plates including a tenocytes with tenocytes tenocytes with MSCs, and tenocytes with Pio-MSCs. After 48 h of co-culturing, tenocytes were harvested and the concentration of collagen in the supernatant was measured using a SirCol assay kit (Biocolor Inc., Carrickfergus, UK). For gene expression analysis, total RNA was extracted using TRIzol, and cDNA was synthesized using Power SYBR Green PCR Master Mix (Thermo Fisher Scientific, Waltham, MA, USA). qRT-PCR was performed using the QuantStudio 6 Flex Real-Time PCR System (Thermo Fisher Scientific, Waltham, MA, USA). The sequences of primers used were as follows: GAPDH, 5-TGACTCTACCCACGGCAAGTTCAA-3 and 5-ACGACATACTCAGCACCAGCATCA-3; Col1, 5′-GAGAGGTGAACAAGGTCCCG-3′ and 5′-AAACCTCTCTCGCCTCTTGC-3′; Col3, 5′-GAGGAATGGGTGGCTATCCG-3′ and 5′-TCGTCCAGGTCTTCCTGACT-3′; Scleraxis (SCX), 5′-AGACCGTGACAGAAAGACGG-3′ and 5′-CTGTTCATAGGCCCTGCTCA-3′; TnC, 5′-CTACCGACGGGATCTTCGAC-3′ and 5′-TTCCGGTTCAGCTTCTGTGG-3′. The change in mRNA expression was determined using the previously described 2^−ΔΔCT^ method [29].

### 4.7. Statistical Analysis

All experiments were performed at least three times. All values are expressed as a mean ± standard error. A Student’s *t*-test or one-way analysis of variance with post hoc analysis of the Tukey multiple comparison tests were performed for intergroup comparison. *p*-values < 0.05 were considered to indicate statistical significance and were calculated using PASW Statistics 18 for Windows (SPSS Inc., Chicago, IL, USA).

## 5. Conclusions

Pioglitazone can be used to enhance the therapeutic effects of MSCs for tendon repair, as shown by the increased proliferation, migration, remolding gene expression, and ECM secretion of tenocytes. Considering the limited therapeutic effect of current tendon repair protocols using MSCs, our method may contribute to the future development of an efficient and safe alternative to tendon remodeling and repair in a clinically-feasible way.

## Figures and Tables

**Figure 1 ijms-20-00472-f001:**
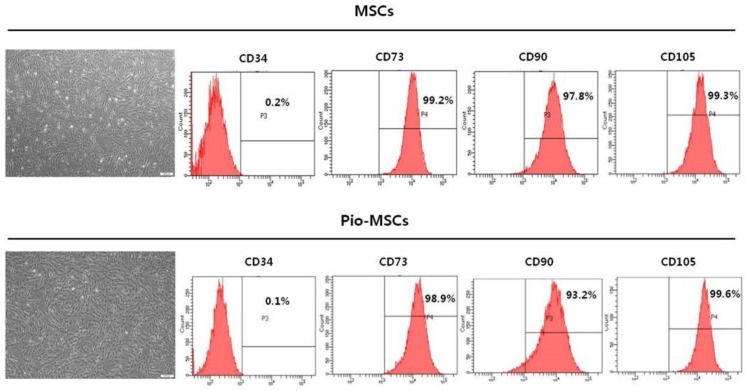
Comparison of the morphology and cell surface marker profile between human Wharton’s jelly mesenchymal stem cells (MSCs) and pioglitazone-pretreated MSCs (Pio-MSCs). Both cell types showed a typical MSC morphology with a spindle- or fibroblast-like appearance. Flow cytometry indicated that both types were positive for the known MSC markers CD73, CD90, and CD105, but negative for CD34.

**Figure 2 ijms-20-00472-f002:**
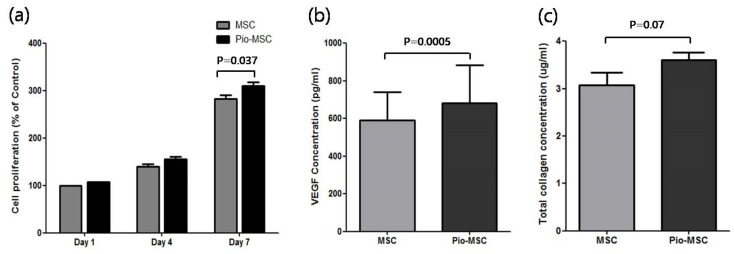
Cell proliferation and relative soluble protein secretion analyses of MSCs and Pio-MSCs. (**a**) Growth profiles were measured in MSCs and pioglitazone-treated MSCs at designated study points. MSCs were cultured in serum free media. MSCs and Pio-MSCs were cultured for 48 h and the concentrations of VEGF (**b**) and collagen (**c**) were measured using ELISA and a SirCol assay, respectively.

**Figure 3 ijms-20-00472-f003:**
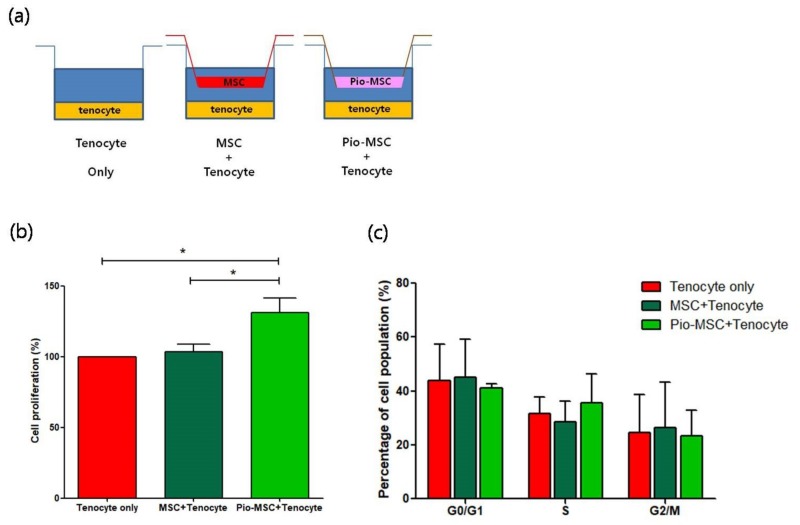
Proliferation profile of tenocytes following co-culture with MSCs or pio-MSCs under an indirect co-culture system. (**a**) Schematic model of the co-culture system of tenocytes with MSCs or pio-MSCs using transwell inserts with a 0.4-μm porous membrane to separate the cells. Each cell type was grown independently on the transwell plates. (**b**) Tenocyte proliferation analysis using an MTT assay. After 48 h in co-culture with MSCs or pio-MSCs, tenocytes were harvested and their proliferation was calculated and normalized against a tenocyte monoculture (tenocyte only). (**c**) At 48 h co-culturing with MSCs or pio-MSCs, the percentage of tenocytes in each phase of the cell cycle was measured by flow cytometry. All data are expressed as a mean ± standard error (SE) from three replicate experiments. * *p* < 0.05.

**Figure 4 ijms-20-00472-f004:**
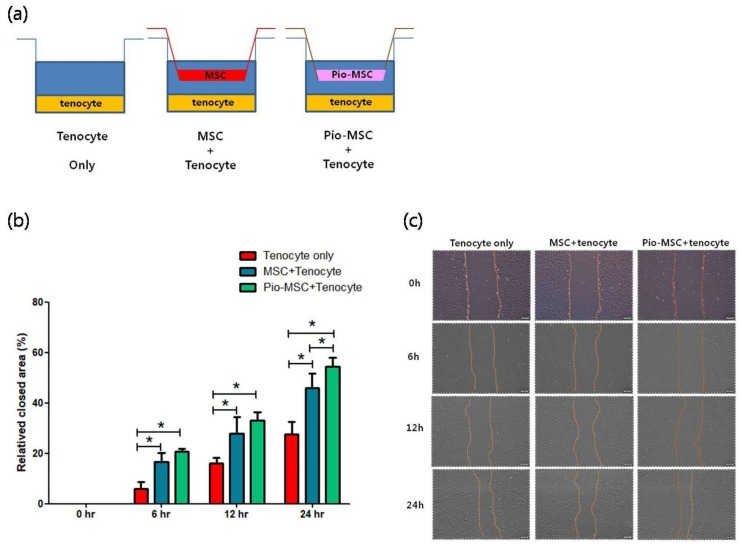
Migration assay of tenocytes following indirect co-culture with MSCs or Pio-MSCs. (**a**) Schematic model of the co-culture system used. Tenocytes were co-cultured with a tenocyte control, and with MSCs and pio-MSCs using transwell inserts with a 0.4-μm porous membrane to separate the cells. Each cell type was grown independently on the transwell plates. (**b**) Relative tenocyte migration area changes following co-culture with MSCs or Pio-MSCs. The migration areas at the designated study points were normalized against that obtained at 0 h. * *p* < 0.05. (**c**) Light microscopy images of tenocyte migration at the designated study points. The migration area was calculated using an inherent protocol in ImageJ software. Scale bars are 200 µm.

**Figure 5 ijms-20-00472-f005:**
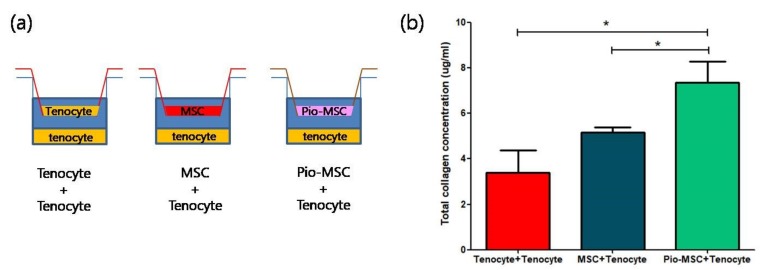
Comparison of relative soluble collagen secretion following co-culture with tenocytes, MSCs, and Pio-MSCs using an indirect transwell system. (**a**) Schematic model of the co-culture system used. (**b**) Tenocytes, MSCs, and Pio-MSCs were each co-cultured with tenocytes for 48 h. The concentration of collagen was then measured using a SirCol assay. The data are expressed as a mean ± standard error (SE) from three replicates. * *p* < 0.05.

**Figure 6 ijms-20-00472-f006:**
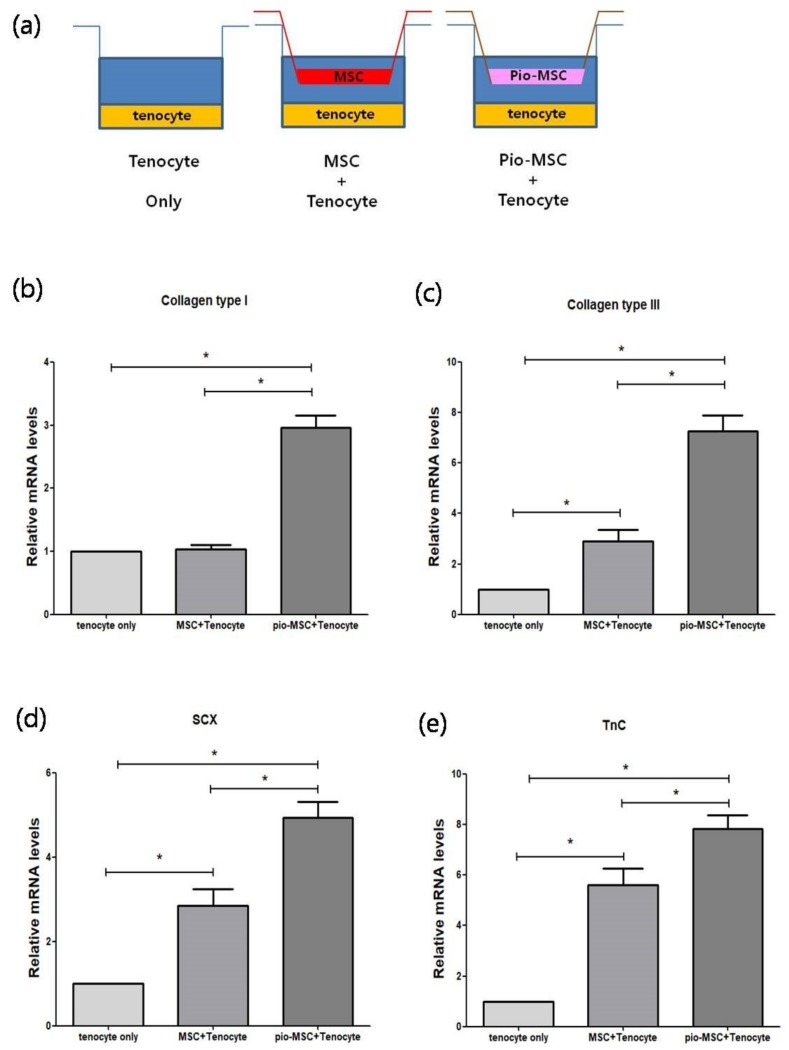
Comparison of the relative mRNA expression levels in co-cultures of tenocytes with MSCs or Pio-MSCs using an indirect transwell system. (**a**) Schematic model of the co-culture system used. Tenocytes were co-cultured with MSCs and Pio-MSCs and then subjected to qRT-PCR analysis for (**b**) collagen Type I, (**c**) collagen type III, (**d**) SCX, and (**e**) TnC. The expression levels in each case were normalized against those in the tenocyte only culture. All data are expressed as a mean ± standard error (SE) from five replicates. * *p* < 0.05.

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
