# Peer review of "Pioglitazone-Primed Mesenchymal Stem Cells Stimulate Cell Proliferation, Collagen Synthesis and Matrix Gene Expression in Tenocytes"

_ijms, 2019, doi:10.3390/ijms20030472_

Round 1
Reviewer 1 Report
It was recently shown that pretreatment of mesenchymal stromal cells (MSCs) with Pioglitazone, a PPAR-gamma agonist, can improve their therapeutic efficacy in animal models of myocardial infarction and lung emphysema. In the submitted study, the Authors investigated the effects of pioglitazone-pretreated MSCs (Pio-MSCs) on tenocytes in vitro. They observed that pioglitazone pretreatment enhanced proliferation, migration, matrix gene expression and collagen secretion of tenocytes, suggesting improved therapeutic efficacy in tendon repair.
The concept underlying this study is derived from previously published work, the methodology is simple and the results look fair. However, the biological significance of these results is uncertain since differences obtained between MSCs and Pio-MSCs are small. Complementing in vitro data with significant effects in vivo could greatly improve the impact of this work.
Author Response
It was recently shown that pretreatment of mesenchymal stromal cells (MSCs) with Pioglitazone, a PPAR-gamma agonist can improve their therapeutic efficacy in animal models of myocardial infarction and lung emphysema. In the submitted study, the Authors investigated the effects of pioglitazone-pretreated MSCs (Pio-MSCs) on tenocytes in vitro. They observed that pioglitazone pretreatment enhanced proliferation, migration, matrix gene expression and collagen secretion of tenocytes, suggesting improved therapeutic efficacy in tendon repair.
Point 1. The concept underlying this study is derived from previously published work, the methodology is simple and the results look fair. However, the biological significance of these results is uncertain since differences obtained between MSCs and Pio-MSCs are small. Complementing in vitro data with significant effects in vivo could greatly improve the impact of this work.
Response 1. We appreciate the reviewer’s comment. As said, our in vitro data may not be sufficient to support the enhanced function of Pio-MSCs on tendon-related pathology per se. However, we consider that our finding in terms of Pio-MSC’s potent role in enhancing proliferation, migration, matrix gene expression and collagen secretion of tenocytes could possibly be applied to developing a novel method to functionally improve the regenerative potential of MSCs in treating tendon diseases. As suggested, we are currently investigating whether Pio-MSCs have better function in tendon recovery using a rat model of tendon injury.
Reviewer 2 Report
Won Kim, Seul Ki Lee, Young-Won Kwon, Sun G. Chung, and Soo Kim reported a manuscript entitled, “Pioglitazone-primed mesenchymal stem cells stimulate cell proliferation, collagen synthesis and matrix gene expression in tenocytes.” to international Journal of Molecular Sciences.
Pioglitazone is one of the currently popular hypoglycemic agents acting on peroxisome proliferator-activated receptor gamma, PPAR-γ, of which activation seems to affect the differentiation of MSCs and activated MSCs, mesenchymal stem cells, when pre-treated with pioglitazone, demonstrate significantly improved therapeutic effects in myocardial infarction and emphysema in animals.
Figure 2B and 2C should clarify which day these were observed.
Figures 3 to 6, the mechanism and mode of action are missing. The authors are strongly recommended to elucidate which in fact affect in favor of Pioglitazone-pretreated MSCs, Pio-MSCs in proliferation, migration, protein expression and transcript upregulations.
Tenocytes are not solely involved in tendon repair in a real world and at least several other cell types or preferably with animal models, this should be clarified with Pio-MSCs.
What will it happen when pioglitazone and MSCs are treated simultaneously?
Author Response
Pioglitazone is one of the currently popular hypoglycemic agents acting on peroxisome proliferator-activated receptor gamma, PPAR-γ, of which activation seems to affect the differentiation of MSCs and activated MSCs, mesenchymal stem cells, when pre-treated with pioglitazone, demonstrate significantly improved therapeutic effects in myocardial infarction and emphysema in animals.
Point 1: Figure 2B and 2C should clarify which day these were observed.
Response 1: Thank you for the suggestion. We have measured the concentration of VEGF and collagen at 48 hours of treatment and it has been described in the legend of Figure 2.
Point 2: Figures 3 to 6, the mechanism and mode of action are missing. The authors are strongly recommended to elucidate which in fact affect in favor of Pioglitazone-pretreated MSCs, Pio-MSCs in proliferation, migration, protein expression and transcript upregulations.
Response 2: We appreciate the reviewer’s comment and agree to it. On the other hand, we also consider that defining MOA of biomedicinal product (i.e. MSCs or its secretome) is currently not easy. Therefore, (further) validation of MOA using in vitro experiments may not show key bioactive components or pathways that led to our results. Alternatively, to better find its MOA, we are now planning to assess its overall efficacy by an animal study.
Point 3: Tenocytes are not solely involved in tendon repair in a real world and at least several other cell types or preferably with animal models, this should be clarified with Pio-MSCs.
Response 3: As commented by the reviewer, tenocytes are not the only cell type that are involved in tendon repair. However, previous studies support our notion that tenocytes are an important player in tendon regrowth [1]. In this regard, we investigated the trophic role of Pio-MSCs on tenocytes, and our data is the first to demonstrate the effect of Pioglitazone on enhancing the function of MSCs. We will conduct animal experiments to finally confirm such role in vivo.
Reference
Riley G, The pathogenesis of tendinopathy. A molecular perspective. Rheumatology (Oxford). 2004;43(2):131-42.].
Point 4: What will it happen when pioglitazone and MSCs are treated simultaneously?
Response 4: Our intention was to investigate the role of pretreatment of Pioglitazone on MSCs to see whether such process can enhance the function of MSCs on tenocyte growth. Accordingly, we removed Pioglitazone from culture medium after 7 days of treatment, followed by co-culture of tenocytes and Pioglitazone treated MSCs. We consider if Pioglitazone and MSCs are treated at the same time, different outcome would be found due to the direct effect of Pioglitazone on tenocytes.
Round 2
Reviewer 1 Report
The Authors should point out in the discussion that further in vivo studies are required to confirm the biological significance of their results, since differences in in vitro activities between MSCs and Pio-MSCs are small. (for instance, they could modify lines 236-237).
Author Response
Point 1: The Authors should point out in the discussion that further in vivo studies are required to confirm the biological significance of their results, since differences in in vitro activities between MSCs and Pio-MSCs are small. (for instance, they could modify lines 236-237).
Response 1: We agree to the reviewer’s comment. We have added the preliminary data on the enhanced function of Pio-MSCs using rat model of tendon injury (supplementary figure 2). Also, we have modified the sentences to better describe the necessity of confirming such effect using animal models (line 237).
Reviewer 2 Report
Won Kim, Seul Ki Lee, Young-Won Kwon, Sun G. Chung, and Soo Kim reported a manuscript entitled, “Pioglitazone-primed mesenchymal stem cells stimulate cell proliferation, collagen synthesis and matrix gene expression in tenocytes.” to international Journal of Molecular Sciences.
The magnitude of using pioglitazone pre-treatment is not greater compared to the MSCs alone, which can be variable from cell property or conditions such as cell medium, even though there were significant differences between the two.
In order properly to understand mechanism, at least an animal study seem mandatory beyond this system proposed.
Author Response
Point 1: The magnitude of using pioglitazone pre-treatment is not greater compared to the MSCs alone, which can be variable from cell property or conditions such as cell medium, even though there were significant differences between the two. In order properly to understand mechanism, at least an animal study seem mandatory beyond this system proposed.
Response 1: We agree to the reviewer’s comment. To circumvent the possible variations that can be occurred from using multiple donors, all data were generated using independent MSCs from three donors. Also, to support the understanding of its mechanism, we have added new preliminary data on the increased function of Pio-MSCs using a rat model of tendon injury.
Round 3
Reviewer 2 Report
Won Kim, Seul Ki Lee, Young-Won Kwon, Sun G. Chung, and Soo Kim reported a manuscript entitled, “Pioglitazone-primed mesenchymal stem cells stimulate cell proliferation, collagen synthesis and matrix gene expression in tenocytes.” to international Journal of Molecular Sciences.
Probably as induction by pre-treatment by pioglitazone-priming to mesenchymal stem cells seem relatively weak, in order to clarify the further detailed cell-to-cell interaction, it may be required to set tenocyte in the upper wells in figures 3 to 6.
Supplementary data seem to explain that AKT or ERK1/2 are indeed involved in the subsequent protein augmentations.
Author Response
Point 1: Probably as induction by pre-treatment by pioglitazone-priming to mesenchymal stem cells seem relatively weak, in order to clarify the further detailed cell-to-cell interaction, it may be required to set tenocyte in the upper wells in figures 3 to 6.
Response 1: To investigate the trophic role of Pio-treated MSCs on tendon regeneration, we designed a co-culture system to mimic the environment where MSCs or Pio-MSCs are transplanted into the tendon tissue. Therefore, we compared the paracrine effect of MSC and Pio-MSC by placing the MSC or Pio-MSC in the upper layer and the tenocytes below. Compared with MSC, Pio-MSC significantly increased the proliferation and their function of tenocytes.
Point 2: Supplementary data seem to explain that AKT or ERK1/2 are indeed involved in the subsequent protein augmentations.
Response 2: Thank you for the appreciation on supplementary data. We will further compare the effect of MSC and Pio-MSC on the regeneration of damaged tendons tissue via animal experiment. In more detail, we will analyze the difference of major signalling pathways to compare the role of Pio-MSC to identify potential MOA, including AKT and ERK1/2.